# The Paradox of Coenzyme Q_10_ in Aging

**DOI:** 10.3390/nu11092221

**Published:** 2019-09-14

**Authors:** M. Elena Díaz-Casado, José L. Quiles, Eliana Barriocanal-Casado, Pilar González-García, Maurizio Battino, Luis C. López, Alfonso Varela-López

**Affiliations:** 1Institute of Biotechnology, Department of Physiology, Biomedical Research Center, University of Granada, Avda del Conocimiento sn, 18016 Granada, Spain; eli_bcoct90@hotmail.com (E.B.-C.); pgonzalez@ugr.es (P.G.-G.); luisca@ugr.es (L.C.L.); 2Centro de Investigación Biomédica en Red de Fragilidad y Envejecimiento Saludable (CIBERFES), 18016 Granada, Spain; 3Institute of Nutrition and Food Technology “José Mataix Verdú”, Department of Physiology, Biomedical Research Center, University of Granada, Avda del Conocimiento sn, 18016 Granada, Spain; jlquiles@ugr.es; 4Department of Clinical Sicences, Università Politecnica delle Marche, 60131 Ancona, Italy; mbattino@mta01.univpm.it; 5Nutrition and Food Science Group, Department of Analytical and Food Chemistry, CITACA, CACTI, University of Vigo, 36310 Vigo, Spain; 6International Research Center for Food Nutrition and Safety, Jiangsu University, Zhenjiang 212013, China

**Keywords:** mitohormesis, antioxidant, mitochondria, anti-aging, diet, aging-related diseases

## Abstract

Coenzyme Q (CoQ) is an essential endogenously synthesized molecule that links different metabolic pathways to mitochondrial energy production thanks to its location in the mitochondrial inner membrane and its redox capacity, which also provide it with the capability to work as an antioxidant. Although defects in CoQ biosynthesis in human and mouse models cause CoQ deficiency syndrome, some animals models with particular defects in the CoQ biosynthetic pathway have shown an increase in life span, a fact that has been attributed to the concept of mitohormesis. Paradoxically, CoQ levels decline in some tissues in human and rodents during aging and coenzyme Q_10_ (CoQ_10_) supplementation has shown benefits as an anti-aging agent, especially under certain conditions associated with increased oxidative stress. Also, CoQ_10_ has shown therapeutic benefits in aging-related disorders, particularly in cardiovascular and metabolic diseases. Thus, we discuss the paradox of health benefits due to a defect in the CoQ biosynthetic pathway or exogenous supplementation of CoQ_10_.

## 1. Introduction

### 1.1. CoQ Biosynthesis

Coenzyme Q (CoQ), ubiquinone or 2,3-dimethoxy-5-methyl-6-polyprenyl-1,4-benzoquinone is a two-part molecule composed of a benzoquinone ring, which has redox active sites, and a long polyisoprenoid lipid chain that positions the molecule in the mid-plane of a membrane bilayer. The length of the chain depends on the species, i.e., 10 isoprene units (CoQ_10_) in humans and *S. pombe*, eight units (CoQ_8_) in *Escherichia coli*, six units (CoQ_6_) in *Saccharomyces cerevisiae* and nine units (CoQ_9_) in rodents. Nevertheless, some species have more than one CoQ form, e.g., human cells and tissues also contain CoQ_9_ and rodent cells and tissues also contain CoQ_10_ as minor forms. 

The synthesis of CoQ in eukaryotes mainly occurs in mitochondria by a set of nuclear-encoded CoQ proteins through a biochemical pathway that has not been fully defined. Moreover, some authors have suggested that CoQ may also be produced outside of mitochondria, mainly in the endoplasmic reticulum-Golgi [1,2]. On the other hand, recent studies suggest that the endoplasmic reticulum mitochondria encounter structure (ERMES) might promote CoQ biosynthesis and the movement of CoQ and its precursors and intermediates between these organelles [3]. Thus, the mitochondrial production of CoQ could contribute to extramitochondrial pools of CoQ. The known steps required for CoQ production were first identified in *S. cerevisiae* and later confirmed in mammals. Specifically, *S. cerevisiae* CoQ (CoQ1–CoQ9) mutants, but also *E. coli* ubi (ubiA-J and ubiX) mutants, have allowed identifying the CoQ genes and biosynthetic intermediaries of the biosynthetic pathway of CoQ [4,5,6,7,8,9]. At least 11 genes (ubiA, B, C, D, E, F, G, H, I, J, and X) in *E. coli* and 13 genes (YAH1, ARH1, and CoQ1–CoQ11) in *S. cerevisiae* are involved in CoQ biosynthesis. Human and mouse orthologues for almost all of these genes have already been identified (with the possible exception of *CoQ11*). In the cases of *CoQ8* and *CoQ10*, humans have two orthologues for each one: *ADCK3* (*CoQ8A*) and *ADCK4* (*CoQ8B*), and *CoQ10A* and *CoQ10B*, respectively. In *Homo sapiens* the polypernyl diphosphate synthase is a heterotetramer formed by the dimer of PDSS1 and PDSS2 proteins [10,11]. The expression of the human homologs (*CoQ1–CoQ10*) for yeast genes restores CoQ_6_ biosynthesis in the corresponding *S. cerevisiae CoQ* mutants, indicating the profound functional conservation in human cells [12].

Structurally, CoQ biosynthesis could be divided into four stages: quinone synthesis, isoprenoid tail synthesis, both molecules condensation and ring modification (Figure 1a).

#### 1.1.1. Quinone Synthesis

The primary aromatic precursor of the benzoquinone ring is 4-hydroxybenzoic acid (4HB). For the biosynthesis of 4HB, human cells utilize phenylalanine or tyrosine as ring precursors. Many of the steps involved in the generation of 4HB from tyrosine are not completely understood and the responsible enzymes are unidentified [13,14]. Current studies have identified the first and last reactions of this pathway and the enzymes involved in these reactions in yeast. Thus, 4HB synthesis starts with the deamination of tyrosine to 4-hydroxyphenylpyruvate by Aro8 and Aro9 and finishes with the oxidation of 4-hydroxybenzaldehyde to 4HB by Hfd1. Human Hdf1 homolog ALDH3A1 plays a role in CoQ biosynthesis and is able to rescue CoQ deficiency in yeast with the inactivated *HFD1* gene [15,16]. Aro8 and Aro9 human homolog remain unidentified, but α-aminoadipate aminotransferase (AADAT) and the tyrosine aminotransferase (TAT) have been postulated as candidates [17]. 

Although 4HB is depicted as the main precursor of CoQ, other aromatic ring precursors may be incorporated into CoQ biosynthesis. The 2,4-dihydroxybenzoic acid (2,4-diHB), 3,4-dihydroxybenzoic acid (3,4-diHB) and vanillic acid are analogs of 4HB that may allow the stimulation and/or bypass of CoQ biosynthesis defects in yeast *CoQ6* and *CoQ7* mutants, as well as in *CoQ7* conditional knockout mice, *CoQ9^R239X^* mice and human skin fibroblasts from patients with mutations in *CoQ7* and *CoQ9* [18,19,20,21,22,23]. Para-coumarate and resveratrol also can be used as CoQ ring precursors in *E. coli*, *S. cerevisiae*, and human and mouse cells [24]. Moreover, in mouse and human kidney cells, kaempferol is able to increase CoQ levels. This phenolic compound apparently acts as a biosynthetic ring precursor competing with 4HB [25].

#### 1.1.2. Polyisoprenyl Tail Synthesis 

The lipophilic polyprenyl tail is synthetized by the addiction of dimethylally pyrophosphate (DMAPP) and isopentenyl pyrophosphate (IPP) to form decaprenyl diphosphate and nonaprenyl diphosphate via PDSS1/PDSS2 in humans and mice. Consequently, the heterotetramer PDSS1/PDSS2 is responsible for the length of the polyprenyl tail. The polyprenyl precursors come from the mevalonate pathway and are generated in the cytosol. The mechanism by which these precursors are transported from the cytosol to mitochondria is unknown, and it has been suggested that some transporters may exist in the inner mitochondrial membrane [17]. 

#### 1.1.3. Attachment of the Ring/Chain

Furthermore, 4HB and polyprenyl tail are condensed in a reaction catalyzed by human or mice CoQ2 to form 3-decaprenyl-4HB and 3-nonaprenyl-4HB (Figure 1a). 

#### 1.1.4. Next Steps of CoQ Biosynthesis-Ring Modifications

Then, 3-decaprenyl-4HB or 3-nonaprenyl-4HB undergoes subsequent modifications of the ring through a set or reactions of methylations, decarboxylation and hydroxylations to finally generate the molecule of CoQ_10_ or CoQ_9_.

The first reaction is the C5-hydroxylation, catalyzed by CoQ6, to produce 3-decaprenyl-4,5-dihydroxybenzoic acid and 3-nonaprenyl-4,5-dihydroxybenzoic acid. In yeast, this step requires two enzymes, ferredoxin Yah1p and ferredoxin reductase Arh1p, which provide electrons for CoQ6 activity. It is unknown whether the mammalian orthologs, ferredoxin reductase (FDXR) and ferredoxin 2 (FDX2), respectively, act in mammalian CoQ biosynthesis. CoQ6 is characterized as a flavin-dependent monooxygenase peripherally associated with inner mitochondrial membrane on the matrix face [26,27]. 

C5-hydroxylation is followed by O-methylation catalyzed by CoQ3 to form 3-decaprenyl-4-hydroxy-5-methoxybenzoic acid and 3-nonaprenyl-4-hydroxy-5-methoxybenzoic acid. CoQ3 is an S-adenosylmethionine-dependent methyltransferase that is involved in two O-methylation reactions in the CoQ biosynthetic pathway. This polypeptide is also peripherally associated with the mitochondrial inner membrane on the matrix face [28,29].

The proteins involved in the subsequent steps, C1-decarboxylation and C1-hydroxylation, have not been identified in the eukaryotes. However, in prokaryotes the decarboxylation and hydroxylation in prokaryotes are catalyzed by the 3-Octaprenyl-4-hydroxybenzoate decarboxylase UbiD and the 2-octaprenyl-6-methoxyphenol hydroxylase UbiH, respectively [30,31]. In prokaryotic decarboxylation, the flavin prenyltransferase UbiX is important for synthesizing prenylated flavin, which is used as a cofactor for UbiD [32]. Nevertheless, there is no sequence homolog for UbiD or UbiX in humans, suggesting that the C1-hydroxylation could be part of the decarboxylation mechanism.

The ring is then further modified by C2-methylation in a reaction catalyzed by CoQ5. In this reaction, 2-decaprenyl-6-methoxy-1,4-benzenediol and 2-nonaprenyl-6-methoxy-1,4-benzenediol is converted to 2-decaprenyl-3methyl-6-methoxy-1,4-benzenediol and 2-nonaprenyl-3methyl-6-methoxy-1,4-benzenediol (DMQH_2_) [33]. CoQ5 is an S-adenosyl methionine (SAM)-dependent methyltransferase (SAM-MTase) that is peripherally associated with the mitochondrial inner membrane on the matrix face [34]. DMQH_2_ is converted to 2-decaprenyl-3methyl-6methoxy-1,4,5-benzenetriol and 2-nonaprenyl-3methyl-6methoxy-1,4,5-benzenetriol (DMeQH_2_) in a C6-hydroxylation catalyzed by CoQ7, which is a carboxylate bridged diiron hydroxylase peripherally associated with the mitochondrial inner membrane on the matrix side [35]. 

The final step of the CoQ biosynthetic pathway is an O-methylation also catalyzed by CoQ3 (Figure 1a). 

#### 1.1.5. Other Mitochondrial Proteins Involved in CoQ Biosynthesis

There are additional proteins needed for CoQ biosynthesis since their dysfunctions induce a decline in the levels of CoQ. However, their molecular exact functions are yet unclear. These are the cases of CoQ8A, CoQ8B, CoQ9, CoQ_10_A, CoQ_10_B and CoQ4. 

CoQ8A (=ADCK3) and CoQ8B (=ADCK4) have been related with CoQ_10_ biosynthesis and mutations in these proteins are associated with CoQ_10_ deficiency [36,37,38,39]. These two proteins are considered the human orthologues of yeast CoQ8p because CoQ_6_ biosynthesis is rescued by the expression of human CoQ8A and CoQQB in CoQ8-mutant yeast strains [39,40]. However, the precise biochemical activity and the role of CoQ8 in CoQ biosynthesis remains unknown. Although initially CoQ8 was hypothesized to be a protein kinase that may phosphorylate other CoQ biosynthetic proteins, recent studies have demonstrated that CoQ8A shows ATPase activity that is essential for CoQ biosynthesis. In any case, it seems that the activity of CoQ8A could stabilize Complex Q [41].

CoQ9 is a lipid-binding protein associated with the inner mitochondrial membrane, on the matrix face [42]. This protein binds aromatic isoprenes with high specificity, including CoQ intermediates that likely reside within the bilayer. Therefore, CoQ9 seems to present the intermediates of the CoQ biosynthetic pathway directly to CoQ enzymes [43]. CoQ9 is also necessary for the stability and activity of CoQ7, so *CoQ9* mutant mice and human cells show reduced levels of CoQ7 and accumulation of DMQH_2_, the substrate of the reaction catalyzed by CoQ7 [23,27,44,45,46]. 

CoQ_10_A and CoQ_10_B in humans [47] are homologues of the yeast CoQ_10_, a protein that contains a lipid-binding domain that binds CoQ and CoQ intermediates, and that is required for efficient biosynthesis of CoQ_6_ and electron transport in the mitochondrial respiratory chain. The molecular basis for these effects is unknown, but CoQ_10_ probably directs to CoQ from synthesis site to function sites [48,49,50].

Finally, the function of the CoQ4 is still unknown, although some studies suggest that it is required for the assembly and stability of Complex Q, as it is associated with CoQ3, CoQ5, CoQ6 and CoQ9 [28,51,52]. Also, it is speculated that CoQ4 could bind to CoQ and CoQ intermediates and it could organize the enzymes that perform the ring modifications. In humans, there are two different isoforms of the CoQ4 polypeptide but only one is localized in mitochondria [53]. 

#### 1.1.6. Complex Q

CoQ biosynthesis depends on a multi-subunit CoQ polypeptide complex formed by several CoQ biosynthetic proteins and by non-proteinaceous components, including some CoQ intermediates and CoQ itself. It is widely accepted that CoQ3–CoQ9 assembly into a high molecular mass complex called Complex Q. These complexes are peripherally associated with the matrix-side of the inner mitochondrial membrane and are essential feature for the de novo biosynthesis of CoQ [17,51,54,55]. In fact, the lack of a single CoQ polypeptide in yeasts, mice and human cells results in the degradation of several CoQ proteins [23,40,42,45,56]. 

The functional role of Complex Q is to enhance the catalytic efficiency, optimizing the orientation of the substrates and active sites of the enzymes, and to minimize the escape of intermediates, which could be toxic due to their redox or electrophilic properties [17,45,54]. However, many questions about Complex Q remain unknown, including its regulation and the key components involved in its formation and stability. 

### 1.2. Functions of CoQ

The lipophilic characteristic of CoQ together with its redox capacity allows this molecule to participate in multiple cellular pathways and functions. 

#### 1.2.1. The Role of CoQ in the Mitochondrial Respiratory Chain

The best-known role of CoQ is its function as an essential electron carrier in the mitochondrial respiratory chain (MRC). CoQ passes electrons between complex I (NADH-ubiquinone oxidoreductase) or Complex II (succinate-ubiquinone oxidoreductase) and Complex III (succinate-cytochrome c oxidoreductase) [57] (Figure 1b). Thus, CoQ can be found in both oxidized (CoQ or ubiquinone) and reduced forms (CoQH_2_ or ubiquinol), and the conversion between these oxidized-reduced states allows it to act as a cofactor of enzymatic reactions transferring electrons to substrates. This oxidation–reduction cycle may occur by a two-step transfer of electrons producing a semiquinone (CoQH·) intermediate.

In mitochondria, CoQ also accepts electrons from sulfide quinone oxidoreductase (SQOR) during sulfide detoxification [58,59,60,61]; proline dehydrogenase 1 (PDH), an enzyme required for proline and arginine metabolism [62]; coline dehydrogenase (CHDH), required for glycine metabolism; mitochondrial glycerol-3-phosphate dehydrogenase (G3PDH), which connects oxidative phosphorylation, fatty acid metabolism and glycolysis [63]; dihydroorotate dehydrogenase (DHOH), an enzyme involved in pyrimidine biosynthesis [64,65,66]; electron transport flavoprotein dehydrogenase (ETFDH), a key enzyme involved in the fatty acid β-oxidation [67]; and hydroxylproline dehydrogenase, involved in the glyoxylate metabolism [68] (Figure 1b). All these processes generate CoQH_2_, which is re-oxidized by complex III. Therefore, CoQ is an excellent redox carrier involved in different cellular processes since it has the ability to sustain continuous oxidation–reduction cycles.

#### 1.2.2. The Antioxidant Capacity

Reactive Oxygen Species (ROS) formed in the cell are able to damage lipids, proteins and DNA. The mitochondrion is the compartment considered to be the major source of ROS production in the cell and, therefore, this organelle is highly susceptible to suffer oxidative damage. To avoid this harmful effect, the mitochondrion possesses antioxidant compounds and systems, being CoQ one of them. CoQ is one of the most powerful endogenously synthesized membrane antioxidants, being present in all membranes. Its antioxidant function efficiently protects lipids from harmful oxidative damage, but also DNA and proteins [69,70]. 

CoQ is able to prevent lipid peroxidation in most subcellular membranes and its effectiveness to inhibit lipid peroxidation depends on its complex interaction during the peroxidation process. CoQH_2_ has the ability to both prevent production and eliminate directly lipid peroxyl radicals (LOO), while vitamin E acts by quenching these radicals. Also, CoQH_2_ regenerates vitamin E from the α-tocopheroxyl radical [71]. CoQ is considered an efficient antioxidant against radicals produced in membranes for its regeneration capacity, its lipid solubility and because of its involvement in the initiation and propagation of lipid peroxidation [72].

CoQ is effective in the prevention of DNA damage as shown by the decreased strand breaks in DNA and other oxidative DNA damage markers in mitochondria [73], which is important for mitochondrial DNA since its oxidative damage is approximately 10-fold higher than nuclear DNA and is not easily reparable. These data are supported by Tomasetti and colleagues [74,75] who showed that CoQ_10_ supplementation inhibits DNA strand break formation and oxidative damage and enhances DNA repair enzyme activities in lymphocytes. Both the antioxidant activity of CoQ_10_ and the role that CoQ_10_ may play in the redox mechanism implicated in the transactivation of DNA repair enzymes could explain the capability of CoQ_10_ to protect DNA from oxidative damage. However, the precise mechanism under these effects remains unclear.

The high efficiency of CoQ as an antioxidant is related to its effective reactivation redox, its ubiquitous distribution, its localization in membranes, its relatively high concentration and the large capacity of the cell to regenerate CoQ at all its locations [76,77].

#### 1.2.3. Other Controversial Functions

It has been suggested that CoQ is one of the compounds influencing the mitochondrial permeability transition pore (PTP). Studies about the regulation of the PTP by ubiquinone analogues show that ubiquinone inhibits the Ca^2+^ dependent PTP opening, which is mediated through a ubiquinone-binding site directly involved in PTP regulation rather than through redox reactions [78]. However, further details about the organization, structure, components and regulation of the PTP are required to understand whether CoQ has a primary function in the PTP. Other mitochondrial components susceptible to being regulated by redox reactions are the uncoupling proteins (UCPs). However, controversial results have been reported regarding to the involvement of CoQ in the regulation of the UCPs [79,80,81,82,83]. 

### 1.3. Pathologic Conditions with Decreased Levels of CoQ

The levels of CoQ are quite stable in cells. However, CoQ levels can be severely reduced in a group of mitochondrial diseases called CoQ deficiencies, which are clinically and genetically heterogeneous disorders characterized by a decrease in the levels of CoQ in tissues or cells. If the deficiency is caused by pathogenic mutations in the genes required for CoQ_10_ biosynthesis, it is classified as primary CoQ_10_ deficiencies. Nevertheless, secondary CoQ_10_ deficiencies are caused by mutations in genes unrelated to CoQ biosynthesis or are derived from other physiological processes or pharmacological treatments. The secondary forms are probably more frequent than the primary defects. 

CoQ deficiencies cause the inhibition of oxidative phosphorylation and ATP production, since CoQ is an essential component of the mitochondrial respiratory chain. Because CoQ has other roles unrelated to ATP production, CoQ deficiency can impair other vital cellular functions, such as sulfide metabolism [58,59,60], and may induce oxidative damage [46,84,85,86].

#### 1.3.1. Primary CoQ_10_ Deficiencies

Primary ubiquinone deficiency is a subset of mitochondrial diseases caused by autosomal recessive mutations in CoQ genes. Ogasahara et al. [87] described the first patients with primary CoQ_10_ deficiency as two sisters with progressive muscle weakness, fatigability and central nervous system dysfunction, learning disability, and seizures in one and cerebellar syndrome in the other [87]. Since then, approximately 200 patients have been described with pathogenic mutations in PDSS1 [88], PDSS2 [89,90], CoQ2 [88,89,91], CoQ6 [89,92], CoQ7 [22], CoQ4 [93,94,95], CoQ5 [96], CoQ8A [36,37,38], CoQ8B [89,97] and CoQ9 [98] genes. The clinical manifestations are very heterogeneous among different genes and among patients with mutations in individual genes. Multiple organs may be affected and lead to different symptomatology; when the central nervous system (CNS) is affected the clinical manifestations are variable, including encephalopathy, seizures, cerebellar ataxia, epilepsy, intellectual disability, hypotonia, dystonia, spasticity; in kidney, the most common manifestations is steroid-resistant nephrotic syndrome (SRNS); myopathy in muscle and hypertrophic cardiomyopathy in heart. Many symptoms are common to other mitochondrial diseases [99]. 

#### 1.3.2. Secondary CoQ_10_ Deficiencies

Secondary CoQ deficiencies may affect any of the multiple CoQ biological functions and its connection to other metabolic pathways [100,101]. Low CoQ levels have been observed in a wide spectrum of diseases, such as oxidative phosphorylation (OXPHOS) diseases due to defects in nDNA-encoded proteins, OXPHOS diseases due to defects in mtDNA-encoded proteins and non-OXPHOS diseases (mitochondrial energy metabolism disorders and other non-mitochondrial diseases). Examples of diseases that may display secondary CoQ deficiencies include mitochondrial myopathies, mitochondrial DNA depletion syndrome, multiple acyl-CoA dehydrogenase deficiency (by mutations in *ETFDH*), ataxia-oculomotor apraxia syndrome (by mutations in *APTX*), cardiofaciocutaneous syndrome (by mutations in *BRAF*), methylmalonic aciduria, GLUT1 deficiency syndrome, mucopolysaccharidosys type III, spinocerebellar ataxia-10 (by mutations in *ANO10*) or multisystem atrophy (MSA) [102,103,104,105,106]. The symptoms are highly dependent on the original pathology and the most common symptoms are muscular and central nervous system manifestations. The exact mechanisms by which CoQ deficiency occurs are still unknown but it has been speculated that the low CoQ values in patients with mitochondrial diseases may be explained by a combination of different mechanisms, such as an incomplete or abnormal respiratory chain that may affect the formation of Complex Q linked to the respiratory chain (resulting in alterations in CoQ biosynthesis); an increase in CoQ degradation as a consequence of the oxidative stress derived from OXPHOS dysfunction; an interference with signaling pathways that potentially regulate CoQ biosynthesis; or a general deterioration of mitochondrial function [55,107].

Other diseases such as liver cirrhosis, phenylketonuria, fibromyalgia and cardiomyopathies also present a decrease in plasma CoQ levels [101,108], although this parameter is not reliable for a diagnostic purpose [109]. The use of statins to treat hypercholesterolemia can cause secondary CoQ deficiency because statins inhibit the 3-hydroxy-3-methyl-glutaryl-coenzyme A reductase (HMGCR) in the mevalonate pathway, which is required to produce farnesyl pyrophosphate, an intermediate in CoQ biosynthesis [110]. Also, a decrease in the levels of the components of Complex Q has recently been related to secondary CoQ deficiency in various mouse models of mitochondrial diseases [111], as well as in muscle and adipose tissue of patients and a mouse model with insulin resistance [112].

## 2. Long-Life Animal Models with CoQ Deficiency: Potential Mechanisms 

The multiple metabolic pathways that require the function of CoQ can be altered during aging. Also, the function of CoQ as an antioxidant may be important in aging since it is widely reported that there is an increase in oxidative stress with aging. Derived from the free radical theory on aging introduced by Harman et al. [113], oxidative stress resistance and longevity are closely related [113]. Later, Harman [114] published an extension of this theory, showing the mitochondria as a primary source and target of ROS [114]. For that reason, it was unexpected and surprising that some models with defects in the CoQ biosynthetic pathway have increased life expectancy. An explanation for such data has been attempted by introducing the concept of mitochondrial hormesis or mitohormesis, which means that mild mitochondrial stress in early stages of life may induces physiological responses and adaptations in the organism, resulting in a longer life span [115]. 

### 2.1. Worm Models

*Caenorhabditis elegans* has been used extensively as a model organism in different fields of research, including aging. C*lk-1* (*CoQ7* homologous)-deficient worms are characterized by an increased life span together with a deregulation and slowing down of a number of physiological processes, developmental processes and behavioral patterns, e.g., embryonic and postembryonic development, reproduction, periods of defecation, swimming and pumping cycles [116,117,118]. Because the *Clk-1* gene encodes demethoxyubiquinone hydrosylase (CoQ7), then *clk-1* mutant worms are deficient in CoQ7 and, therefore, they cannot produce CoQ_9_ and consequently, accumulate large quantities of demethoxyubiquinone (DMQ_9_) [119]. 

The mechanism underlying the increased longevity in *clk-1* mutants is not clear. Different hypotheses have been proposed. Initially, it was proposed that the low metabolic rate of the *clk-1* mutants could result in a slower production of ROS, which could explain the prolonged life span in these worms [120,121]. Low levels of ROS on the mutants may affect the *ras* pathway and reduce the levels of oxidized LDL-like lipoproteins. These changes may alter the oxidative modifications of cellular constituents in *clk-1* mutants during germline development [122]. However, Braeckman and colleagues [123] did not find significant changes either in metabolic rate or in the activity of two antioxidant enzymes, superoxide dismutase and catalase, in *clk-1* mutants. Also, the same authors reported a mild reduction in oxygen consumption rates and elevated ATP levels in *clk-1* mutants, suggesting an increase in the efficiency of the OXPHOS system [123,124,125,126]. 

Later, it was hypothesized that the *clk-1* mutants prolong life span by a decrease in CoQ levels whether this is directly due to a *clk-1* mutation or a diet lacking CoQ [127]. Also, Asencio and colleagues [128] showed that repressing different CoQ genes in *C. elegans* also extends life span [128], suggesting that the extension of life span could be related to the reduction in the levels of CoQ, rather than to a specific CoQ gene or to the accumulation of DMQ. The authors attributed the extension of life span to the maintenance of the efficiency of respiration together with a reduction in the production of superoxide anion. Nevertheless, Kayser and colleagues [129] showed a profound defect in complex I+III activity in *clk*-1 *C. elegans* mutants, a fact that may contribute to ROS production and its link to life span. Also, we need to take into account that *clk-1* mutant worms obtain significant amounts of CoQ_8_ from diet, which would help to stabilize complex III. In fact, when *clk-1* mutants are fed with *E. coli* mutants that lack CoQ, they show developmental arrest and sterility [119,130], while *clk-1* worms fed with genetically engineered bacteria that produce CoQ_6_, CoQ_7_, CoQ_8_, CoQ_9_ or CoQ_10_ show increased life span, decreased mitochondrial respiration and decreased oxidative damage [118,131]. Similarly, mutants in *mev-1*, a gene that encodes cytochrome b, show elevated superoxide anion production in mitochondria and the exogenous administration of CoQ_10_ reduced superoxide anion levels and extended life span [132]. In addition, exogenous CoQ_10_ supply partially restores the expression of a cluster of genes important for growth and oxidation reactions, which are down and up regulation, respectively, in *clk-1* mutants [133].

On the other hand, DMQ could be a non-functional competitor of CoQ, because it could bind the Q sites in the mitochondrial complexes but it cannot functionally replace CoQ_9_ [118,129,134]. In fact, *clk-1* mutants cannot grow on bacteria producing only DMQ and those fed with a CoQ-less diet die as larvae [130,134]. However, another assay showed that DMQ_9_ would support a sufficient level of respiration and support respiration in later development [129]. The antioxidant properties of DMQ has also been proposed to be responsible for the extension of life span in *clk-1* mutants, although studies in yeast have showed that DMQ is a less effective antioxidant than CoQ [135,136]. Probably, rhodoquinone (RQ_9_) and bacterial CoQ_8_ may also have an influence on life span [127]. All these data support an important role of the CoQ in the aging process in *C. elegans*, minimum levels of CoQ being necessary for adequate embryo development and fertility.

In summary, the reduction in the electron flow on the respiratory chain in *clk-1* mutants worms may explain the low levels of ROS, a physiological adaptation and the increase in life span [127,129] (Figure 2a). Furthermore, the reduced respiration of *clk-1* mutants suggests that the reduction in mitochondrial function determines longevity by a defect specific to complex I-dependent substrates.

### 2.2. Mouse Models

The relationship between a dysfunctional CoQ7 protein or the reduction in the CoQ levels from the embryonic development and the extension of life span has also been observed in two different mouse models, as reported in heterozygous *mclk1* (=*CoQ7*) mice [137] and *CoQ9^Q95X^* mice [138]. 

Heterozygous *mclk1* mice have some common characteristics with *clk-1* mutant worms. Although these mice have normal CoQ levels in most of the tissues, they show a reduced level of MCLK1 protein (approximately 2-fold), reduced ROS levels, decreased oxidative stress and decreased oxidative damage, and their life span is 15–30% longer than wild-type mice [137,139,140].

In livers of old *mclk1^+^*^/*−*^ mice, a loss of *mclk1* expression in most of the cells of the entire hepatic lobules was observed. These cells are more resistant to age-dependent oxidative stress and apoptosis, and they can be propagated in old livers. Presumably, for this reason, the CoQ levels in the liver of old *mclk1^+^*^/*−*^ mice are lower than in the liver of old *mclk1^+^*^/*+*^ mice. However, DMQ is not detectable in any organ of *mclk1^+^*^/*−*^ mice, in contrast to what was observed in worms, suggesting that the increased life span of these mice cannot be due to the presence of DMQ. Moreover, the levels of CoQ in the tissues of young *mclk1^+^*^/*−*^ mice are similar to those in the tissues of *mclk1^+^*^/*+*^ mice. Therefore, these results suggest that the reduction in the levels of MCLK1 and/or the mild decrease in CoQ levels in the liver of old *mclk1^+^*^/*−*^ mice could be responsible for the increased resistance to hepatic oxidative stress and, consequently, for the increased life span in *mclk1^+^*^/*−*^ mice [137,140]. Similar connections have been found in others genetic models, e.g., the *p66sh^−^*^/*−*^ mouse model, the *igf1r^+^*^/*−*^ mouse model and the long-lived dwarf mouse model. In all these cases, a relationship between an increased resistance to oxidative stress and an extension of life span has been identified [141,142,143,144]. 

More recent studies have suggested that CLK-1/MCLK1 could have other roles in mitochondrial function and that CoQ levels are not totally responsible for the extension of life span. Lapointe and Hekimi [145] showed that the reduction in MCLK1 expression does not affect the levels of CoQ in young *mclk1^+^*^/*−*^ mice. However, the electron transport chain, ATP synthesis and the total nicotinamide adenine dinucleotide pool were reduced, and the tricarboxylic acid cycle enzymes were altered. These changes lead to early mitochondrial dysfunction and an increase in mitochondrial oxidative stress, but a reduction in the levels of cytosolic oxidative damage and in the levels of systemic biomarkers of aging [145]. From those data, the authors suggested that although the *mclk1^−^*^/*−*^ clones have only been found in old mice, the anti-aging effects propitiated by low MCLK1 levels already starts at a young age. These data support the existence of a link between aging and mitochondrial energy metabolism, but they seem to be incompatible with the mitochondrial oxidative stress theory of aging. Studies in the long-lived *C. elegans daf-2* mutants, in different *Drosophila* lines and in *C. elegans* under caloric restriction have also found a relationship between elevated mitochondrial ROS production in early stages of life and increase in life span [146,147,148]. This link could be explained by the fact that the early increase in the mitochondrial oxidative stress in the liver induces a reduction in oxidative stress in other cellular compartments, then reducing the systemic oxidative damage, which contributes to the increase in life span. The low levels of oxidized nicotinamide adenine dinucleotide (NAD^+^) would condition a reduced activity of the NADH/NADPH oxidases, the main generators of extra-mitochondrial ROS [140]. Thus, this hypothesis would be compatible with the concept of mitohormesis [33,115] (Figure 2a). 

The *CoQ9^Q95X^* mouse model has a mutation in the *CoQ9* gene that causes a premature termination in the CoQ9 protein. As a result, *CoQ9^Q95X^* mice have undetectable levels of CoQ9 protein and a reduction (approximately 50%) in the levels of CoQ_9_ and CoQ_10_ in kidneys, brain, skeletal muscle and heart, leading to a very mild late-onset mitochondrial myopathy, especially evident in females, and to an increase of 15% in life span [23,46,138]. Because the CoQ9 protein is required for the stability and activity of CoQ7, the lack of CoQ9 in *CoQ9^Q95X^* mice determines a severe reduction in the CoQ7 protein levels in brain, kidney, skeletal muscle, heart and liver. These alterations lead to an impairment in mitochondrial bioenergetic function that is characterized by a reduction in mitochondrial complex I+III activity and oxidative phosphorylation phosphorylating respiration in both, muscle and kidney [23,46,138]. Unlike *mclk1^+^*^/*−*^ mice, *CoQ9^Q95X^* mice have neither a decrease in the CoQ levels in liver, DMQ accumulation nor mitochondrial dysfunction in this tissue. Therefore, there is no evidence that the increased life span in *CoQ9^Q95X^* mice is due to early mitochondrial dysfunction in liver, as proposed in *mclk1^+^*^/*−*^ mice. However, the increased life span in *CoQ9^Q95X^* mice could be attributed to mitochondrial dysfunction in other tissues, such as brain, kidney or muscle, or to unknown mechanisms [138].

## 3. Dietary Supplementation of CoQ_10_ during Aging 

### 3.1. Changes in CoQ Biosynthesis during Aging

The levels of CoQ are quite stable in cells, but its concentration varies among different tissues and organs. Also, the amount of CoQ varies depending on dietary conditions, as the cells can incorporate CoQ from dietary sources [149], and age. During the aging process, as well as in some particular aging-related diseases, a significant reduction in the rate of CoQ biosynthesis seems to occur. Beyer et al. [150] reported decreased CoQ levels in heart, kidney, *gastrocnemius* and oblique muscles in rats at 25 months of age, although CoQ levels in liver increased life span and other tissues like brain and lung had constant CoQ levels [150]. The decrease in CoQ levels with aging has also been reported in some human tissues, where the highest values have been found at 20 years of age [151]. Contrary to those data, Battino and colleagues [152] showed a direct correlation on CoQ and aging in three brain areas of rats [153]. The results showing a reduction in CoQ levels in some tissues during aging would support potential benefits of its exogenous supplementation during aging (Figure 2b). 

### 3.2. CoQ_10_ Supplementation in Aging: Effects on Life span and Longevity

The possibility of increasing CoQ_10_ levels in different organs or tissues through dietary supplementation has been widely explored in recent decades. Studies in rodents [153,154,155,156,157] suggest that CoQ_10_ administration is able to increase the amounts of CoQ_10_ in plasma and liver significantly, and in heart, kidney and skeletal muscle moderately. Similarly, different authors have reported increased systemic levels of CoQ_10_ in humans after supplementing with CoQ_10_ at different daily doses (100 to 2400 mg) and duration (20 days, 3 or even 16 months) in multiple trials [158,159,160,161,162]. Regarding the safety of CoQ_10_ supplements, different assessments in human and animals (reviewed by Hidaka et al. [163]) concluded that the endogenous biosynthesis of CoQ_10_ is not influenced by exogenous inputs. Moreover, it does not accumulate into plasma or tissues when supplementation ends. Based on the absence of adverse effects, the use of a daily dosage of 12 mg/kg of body weight per day in rats following a long-term feeding regimen has been suggested. In humans, a safety level of 1200 mg/day per person has been proposed, although doses of up to 3000 mg/day do not cause serious adverse effects and have been used in shorter clinical trials [164]. Despite no serious adverse effects being found, moderate adverse effects such as nausea and other adverse gastrointestinal effects have been reported. However, these effects were not causally related to the active ingredient because there was no dose–response relationship [163]. More recently, it has been reported that CoQ_10_ is generally safe and well-tolerated at a dose of 2400 mg/day in patients suffering from early-stage Huntington disease [160].

From a biochemical standpoint, CoQ_10_ benefits in relation to aging have been traditionally attributed to their antioxidant properties and to its role in MRC, which would influence mitochondrial functionality and ROS production. Supporting the protective role of CoQ_10_ against oxidative stress, some studies in animals have indicated that CoQ_10_ supplementation can reduce oxidative damage accumulation in certain tissues at least during some stages of the life [156,165]. In mice, 3 weeks of CoQ_10_ supplementation at a dose of 2.81 mg/g of diet was able to attenuate oxidative damage to proteins in liver in aged mice [165]. A similar effect on protein oxidative damage was found in skeletal muscle mitochondria of 14-week-old male rats after 13 weeks of CoQ_10_ supplementation. In the same study, a reductive shift was found in plasma aminothiol status [156]. This could imply an increase in the activity of antioxidant enzymes, higher levels of ROS scavengers or a decreased production of ROS. In humans, supplementation with high amounts of CoQ_10_ (1200 mg/day) to 65 patients undergoing hemodialysis has been shown to reduce plasma levels of the lipid peroxidative damage marker F_2_-isoprostane, which correlates with an increase in plasma CoQ_10_ concentrations [166]. On the contrary, in 55-year-old men, peroxidative markers and total thiols or total antioxidant capacity were also not modified by administrating CoQ_10_H_2_ in spite of the fact that the plasma levels of both total CoQ_10_ and CoQ_10_H_2_ were increased and a higher concentration was reached—notwithstanding that the treatment only had a duration of two weeks [162]. In addition, many of the mentioned studies on animals found effects only in some organs, cell types or organelles; despite some benefits against aging, detrimental effects on health or organ function were observed [156,165]. This might be explained by the different capacity of uptake of exogenous CoQ_10_ of the tissues [167].

In cases in which treatments with CoQ_10_ supplements have been able to reverse or delay some age-related changes, health improvement that would result in lower mortality rates and and extension of life span is expected. The potential of CoQ_10_ supplementation in increasing life span and longevity has been evaluated in different models (Table 1). In C. *elegans*, exogenous CoQ_10_ prolonged life span [168]. Also, the addition of CoQ_10_ to the diet has been shown to increase life span in rodents, at least under certain circumstances—notwithstanding that in most of the studies in both rats and mice, CoQ_10_ supplementation was ineffective for increasing life span [157,169,170]. Importantly, according to the variety of used CoQ_10_ doses (ranged from 10 to 370 mg/Kg per day) and the duration of the different studies, the lack of effects on longevity seems, in many of these studies, not to depend on these conditions. Instead, CoQ_10_ supplementation would be effective in increasing median life span when it is combined with certain nutritional conditions associated with elevated oxidative stress and age-related detrimental effects. From this standpoint, a study on rats comparing CoQ_10_ effects between isocaloric diets with different lipid profiles by using virgin olive, sunflower oil or fish oil as a dietary fat source is particularly interesting [171]. In this study, supplementation of the fat with a low-dose of CoQ_10_ from weaning was able to improve survival in rats receiving a diet with sunflower oil, increasing median life span values. However, no effects were observed in those fed on diets based on virgin olive or fish oil. These effects were observed when sunflower oil was administered in a proportion of 8% *w*/*w* in the diet, which is the double of current recommendations for rodents [172]. This is in concordance with other findings in different tissues and organs of old and young rats fed on similar diets [173,174,175,176,177,178,179,180,181,182,183,184]. Alternatively, it has been suggested that CoQ_10_ supplementation does not truly extend life span, but that it could prevent life span shortening due to oxidative insults [161], as has been suggested by its effect in all aspects related to mitochondrial function, oxidative stress and antioxidant defenses both in animals and humans (Figure 2b). On the other hand, the different oils used could modulate CoQ_10_ absorption since fatty acids significantly help in the absorption of CoQ_10_ via bile acid production [185,186]. In animals fed on similar CoQ-supplemented diets, plasma CoQ_10_ concentration values were 39.24 ± 9.25 and 50.71 ± 5.22 μM when they received a diet based on virgin olive oil and sunflower oil, respectively [174]. Another study reported even lower values in those animals fed on a fish oil-based diet [182]. This seems in concordance with the proposed hypothesis, but the values found in animals receiving different oils were not statistically compared in these studies. 

### 3.3. Reversal of Age-Related Changes by CoQ_10_ Supplementation

The modification of CoQ_10_ levels in different tissues as a consequence of increasing dietary CoQ_10_ intake would be on the basis of the improvement in conditions related to aging observed in both humans and animals. These include obesity and metabolic disorders [158,189,190,191,192], cardiovascular diseases [193,194] or skin aging by exposure to sun radiation. In this sense, dietary CoQ_10_ has been able to reduce insulin resistance (IR) in adults with prediabetes [158], as well as overweight and obese patients with coronary heart disease and type 2 diabetes mellitus (DM2) [190]—notwithstanding that data confirming the effect of CoQ_10_ treatment on blood levels were only provided in the first case. In the last group, the treatment also reduced the serum insulin levels and β-cell function [190], although it did not affect insulin levels in patients with prediabetes [158]. Likewise, it has been reported that the consumption of a Mediterranean diet for 4 weeks led to improvement in parameters related to the mentioned metabolic alterations in old persons, including a reduction in postprandial levels of advanced glycation end products (AGEs) and an increase in AGE receptor-1 and glyoxalase I gene expression. Such effects were accentuated by CoQ_10_ supplementation, which correlated with increased fasting and postprandial CoQ levels in plasma [195]. Also, CoQ_10_ intake during 2 months led to a significant reduction in serum protein oxidative damage markers in type 2 diabetic patients with coronary heart disease, although it did not affect the systemic levels of lipid peroxidative damage markers or thiol concentrations [196]. Still, there were no available data on CoQ levels after the treatment. The therapeutic effects of CoQ_10_ supplementation against metabolic disorders have been confirmed by a meta-analysis on 14 randomized-controlled trials (RCTs) examining CoQ_10_ effects on fasting blood glucose, fasting insulin and HbA1c. This study indicated that CoQ_10_ supplementation slightly but significantly reduced fasting blood glucose, although it does not support the existence of significant effects on fasting insulin and glycosylated hemoglobin [189]. Moreover, results from animal studies suggest that dietary CoQ_10_ would prevent or mitigate diabetes complications [197,198]. In relation to other metabolic alterations, supplementation with CoQ_10_ has also shown beneficial effects on the treatment of hypercholesterolemia and hypertriglyceridemia by modifying blood lipid concentration. In atherosclerosis-prone apolipoprotein E (ApoE) knockout (*Apoe^−^*^/*−*^) mice, which is a well-established mouse model for the study of human atherosclerosis, CoQ_10_ had an anti-atherogenic effect [193]. Supplementation with the reduced form of CoQ_10_ (CoQ_10_H_2_) modulated adipocyte differentiation in female KKAy mice, a model of obesity and DM2, reducing white adipose tissue content and improving brown adipose tissue function. Moreover, the expression of lipid metabolism-related factors was modified, indicating that it regulated lipid metabolism. Among other effects, the decomposition of lipids was enhanced, and the de novo synthesis of fatty acids was inhibited. Thus, the development and progression of obesity could be stopped or at least slowed down [192]. Mechanistically, it has been reported that a seven-day treatment with CoQ_10_H_2_ (250 mg/kg of body weight per day) was able to induce changes in the hepatic expression of lipid metabolism genes functionally connected by the peroxisome proliferator-activated receptor (PPAR)α signaling pathway in C57BL6J mice [199]. However, a meta-analysis of RCTs evaluating the effect of treatment with CoQ_10_ on DM2 patients did not find clear evidence of the capability of CoQ_10_ to alter low-density lipoprotein cholesterol (LDL-C), high-density lipoprotein cholesterol (HDL-C) and blood pressure, although it reduced triacylglyceride (TAG) levels. Moreover, evidence of improved glycemic control by this molecule was not enough [191]. Important conclusions arise from a systematic review compiling RCTs in healthy adults or those at high risk of cardiovascular disease investigating the effects of CoQ_10_ alone as a single supplement in the absence of lifestyle intervention. This systemic review does not support the efficacy of CoQ_10_ supplements in reducing systolic blood pressure, total cholesterol, LDL- C, or HDL-C [200]. Some of the reported effects on systemic markers and metabolic alterations might contribute to cardiovascular disease prevention. In this sense, dietary CoQ_10_ combined with a Mediterranean diet in elderly patients led to increased plasma levels and improved several markers of endothelial function that is also a known risk factor for important cardiovascular diseases [201]. Additional cardiovascular benefits have been found in patients suffering from heart failure, where a short-term (12 weeks or less) daily treatment with oral CoQ_10_ (100 mg per person) improved left ventricular ejection fraction [202]. In contrast, no effect was observed on exercise capacity [203].

CoQ_10_ therapy has also been tested in non-metabolic or cardiovascular diseases. A promising protective effect of ubiquinol (600 mg/kg of body weight per day of ubiquinol for 4 weeks) for kidneys in three-week-old heminephrectomized male Sprague–Dawley rats fed on high salt (8%) diets has been reported [204]. However, a meta-analysis revealed that CoQ_10_ did not have reliable effects against the initiation and progression of diabetic kidney disease [205]. Lastly, the beneficial effects of CoQ_10_ on skin in middle-aged healthy women with Fitzpatrick skin phototypes II and III, where dietary treatment with a water-soluble form of CoQ_10_ limited the seasonal deterioration of viscoelasticity and reduced some visible signs of aging, has been reported [206]. In patients with Alzheimer disease, no changes in F_2_ isoprostanes—neither in biomarkers of amyloid or tau pathology in the cerebrospinal fluid—has been reported after supplementation with 400 mg of CoQ_10_, three times a day for 16 weeks [207]. However, systemic or cerebral CoQ levels were not assessed, so this lack of effects might be related to the previously reported low uptake of CoQ_10_ by the brain [167]. There are also studies supporting the idea that CoQ_10_ exerts anti-apoptotic and anti-inflammatories activities, which could be a consequence of redox-dependent mechanisms. A three-month treatment with CoQ_10_ supplements reduced calcitonin gene-related peptide and tumor necrosis factor (TNF)-α in middle-aged women, although there were no significant differences in serum IL-6 and IL-10 [159]. However, CoQ_10_ intake during two months led to a significant increase in CoQ_10_ plasma levels and a reduction in IL-6 in DM2 patients with coronary heart disease, although no data on CoQ_10_ levels were provided [196]. A meta-analysis of studies in patients with cardio cerebral vascular disease, multiple sclerosis, obesity, renal failure, rheumatoid arthritis, diabetes, and fatty liver disease supports the efficacy of CoQ_10_ in doses ranging from 60 to 500 mg/day, as they showed a decrease in plasma levels of CRP, IL-6 and TNF-α [208], which seems to confirm the anti-inflammatory potential of CoQ_10_ supplements, at least under certain conditions. Likewise, an earlier meta-analysis found that CoQ_10_ in doses ranging from 60 to 300 mg/day significantly reduced the levels of IL-6, although not of CRP [209]. 

Chronic inflammation and oxidative stress play essential roles in the pathogenesis of many age-related diseases in which low CoQ levels may be a pathophysiological factor. Thus, the reported health benefits of CoQ_10_ in previous studies could be a consequence of homeostasis recovery by directly increasing CoQ_10_ or indirectly improving cell antioxidant defenses and/or mitochondrial function. In addition, CoQ_10_ might be useful for good management of the aging process or by preventing age-related alterations that finally lead to disease and death. In this sense, many researchers have investigated the possible effects of long-term interventions with dietary CoQ_10_ on the senescence process itself at different levels (Table 2). Long-life dietary supplementation with CoQ_10_H_2_ slows aging in the senescence-accelerated mouse prone 1 (SAMP1) model in different studies [210,211]. The mechanism of action under this effect has been associated with the deceleration of the normal decline in the expression of *Sirt1, Sirt3, Pgc-1a,* and *Ppara* genes [210]. In contrast, oxidized CoQ_10_ had no effect on senescence in the same model [211]. On the other hand, Sohal et al. [157] did not find effects on the activity of different antioxidant enzymes in the liver, heart, kidney, skeletal muscle, and brain of mice. In male Wistar rats, long-term supplementation with low doses of CoQ_10_ has been shown to attenuate age-related alveolar bone loss [173,212], prevent age-related decline in BMD [174] and reduce histological alterations in endocrine pancreas, which mainly affected β-cell mass and insulin levels in aged animals [175]. In parallel, an aging-associated increase in urinary F2-isoprostanes was prevented by the addition of CoQ_10_, which suggests that CoQ benefits could be a consequence of a reduction in oxidative stress [174]. In gums, an age-related increase in the expression of genes involved in mitochondrial biogenesis and antioxidantdefense has also been found [173]. Such changes could also occur in other tissues, contributing to reducing oxidative stress. However, many of these effects were present only when animals were fed on diets using sunflower oil as a unique dietary fat source. The protective effects of CoQ_10_ have also been observed when this molecule was added to similar diets but with a high-fat content (8% *w*/*w*) [176,177,178,179,180,181,213]. In contrast, the effects of CoQ_10_ on animals receiving diets with other dietary fats as unique fat sources were not so clear. In animals fed a fish oil-based diet, CoQ_10_ has shown a clear benefit on bone health. Namely, animals supplemented with CoQ_10_ showed a higher value of BMD than their younger counterparts, which correlates with lower levels of urinary F_2_-isoprostanes and DNA strand breaks [182]. However, this molecule had no effect on periodontal tissues [173] and pancreas alterations found in aged animals that mainly affected the exocrine gland [175]. The lack of effect exerted by CoQ_10_ was accentuated when it was added to a diet using virgin olive oil since there were practically no changes in markers related to aging in different tissues and organs or improvements in health [173,174].

Overall, studies comparing the effects of the different diets with different fatty acid profiles or the prevalence of a particular dietary fat on aging or certain age-related diseases support that n-6 polyunsaturated fatty acids (PUFA) would be detrimental for health. This negative effect has usually been related to oxidative stress situations or pro-inflammatory environments [214,215]. Thus, it seems that CoQ_10_ is useful to counteract the consequences of unhealthy diets that “accelerate” the aging process, but it has no additional effects under more favorable conditions. In this sense, positive effects against health detriments and aging have been reported for short-term supplementation with CoQ_10_ in animals fed on different high-fat diets [216,217,218]. CoQ_10_ prevented methemoglobin formation and CoQ oxidation induced by feeding young Wistar rats on a high-fat diet for four weeks [216]. CoQ_10_ decreased inflammation and metabolic stress markers in mice fed on a high-fat diet (72%) consisting of corn oil and lard and fructose during 8 weeks [217]. However, the treatment failed in decreasing obesity and tissue lipid peroxides [217]. Likewise, in another study on rats with hyperlipidemia, with insulin resistance and non-alcoholic fatty liver disease (NAFLD) induced by a 4-week high-fat diet (57% of energy), CoQ did not ameliorate the effects of the diet [219]. Similarly, in rats with hyperlipidemia, insulin resistance and NAFLD induced by a 10-week high-fat diet (57% energy from fat), plasma levels of insulin, alanine aminotransferase and HOMA-IR, TG, VLDL and LDL increased and liver lipid accumulation and TG levels were not ameliorated or prevented [219,220]—notwithstanding that apolipoprotein B mRNA and microsomal TG levels were increased, and the phospholipid content of microsomal membranes was altered. In the same sense, CoQ_10_ effects seem to depend on the presence of risk factors or conditions in many cases related to inadequate diets. In that sense, CoQ_10_ reduced protein carbonyl levels and IL6 serum levels [196] and increased plasma glutathione in diabetic patients with coronary heart disease [190]. Other studies in diabetic patients presenting nephropathy suggested that these effects would be mediated, at least in part, by gene expression changes, since CoQ_10_ administration upregulated the gene expression of *PPAR-γ* and downregulated the gene expression of *IL-1* and *TNF-α* in PBMCs [221]. 

Finally, there are also some studies supporting the possible use of CoQ_10_ against the decline in reproductive success with increasing age (i.e., reproductive senescence) and particularly against fertility decline. Female reproductive capacity declines with age as a consequence of an age-related decrease in oocyte quality and quantity, which is accompanied by mitochondrial dysfunction associated with decreased oxidative phosphorylation and reduced ATP level [222,223,224]. Moreover, there is an increase in meiotic spindle abnormalities favoring oocyte aneuploidy that leads to a reduced embryo quality, as well as an increased incidence of miscarriages and birth defects [225]. That the impaired mitochondrial performance associated with reproductive senescence could be a consequence of suboptimal CoQ_10_ availability, which will lead to age-associated oocyte deficits, has been proposed [223]. Actually, a diminished expression of genes encoding two enzymes responsible for CoQ production, Pdss2 and CoQ6, in oocytes of older females in both mouse and human has been reported [223]. Likewise, oocyte-specific Pdss2-deficient animals have also shown a diminished ovarian reserve, a condition prevented by maternal dietary administration of CoQ_10_ [223]. A similar intervention also reversed age-related decline in oocyte quality and quantity in an aging mouse model [223]. In addition, cumulus granulosa cells also display a decline in cell function linked to mitochondrial activity accompanied by a decreased expression of *Pdss2* and *CoQ6* genes in both human and mouse models [225]. Concerning these cells, supplementation with CoQ_10_ restored cumulus cell number, stimulated glucose uptake, and increased progesterone production in an aged mouse model. These preclinical studies suggest that CoQ_10_ might improve fertility in females of advanced maternal age by improving mitochondrial metabolism in oocyte and cumulus cells, which would result in increased quantity and quality [225]. However, there are no clinical studies in aged women confirming this. On the other hand, male infertility has also been associated with oxidative stress and healthy sperm has been correlated with CoQ_10_ levels in seminal fluid [226]. In addition, a meta-analysis of 15 randomized clinical trials suggests that some dietary CoQ_10_ could beneficially modulate sperm quality parameters, improving male fertility. Namely, these interventions have been shown to increase total sperm concentrations and sperm count, and enhanced sperm total motility was enhanced by CoQ_10_ [227]. Therefore, dietary CoQ_10_ could also be useful in reducing male fertility decline. However, the limited sample size of the studies included in the meta-analysis and the inter-study heterogeneity are important factors for carefully interpreting these results [227].

## 4. Conclusions

Some defects in CoQ biosynthesis induce an increase in life span in animal models, most likely due to either a mild reduction in the flow of electrons in the mitochondrial respiratory chain and the subsequent decrease in ROS production or early mitochondrial dysfunction and physiological adaptation to this condition—the latter phenomenon being compatible with the concept of mitohormesis. Moreover, CoQ_10_ supplementation has also shown therapeutic benefits in animal models of disease and human studies, especially in conditions associated with oxidative stress. Those benefits are dependent on CoQ_10_ bioavailability and tissue uptake and, consequently, liver, adipose tissue and circulating cells have a good response to CoQ_10_ supplementation. Therefore, both CoQ biosynthesis defects and CoQ_10_ supplementation are therapeutically relevant depending on the moment and the context of the intervention. However, further studies are required to understand the mechanisms of CoQ_10_ therapy in aging-related diseases, especially in the context of the multiple biological functions of CoQ—an essential molecule for mammalian cells.

## Figures and Tables

**Figure 1 nutrients-11-02221-f001:**
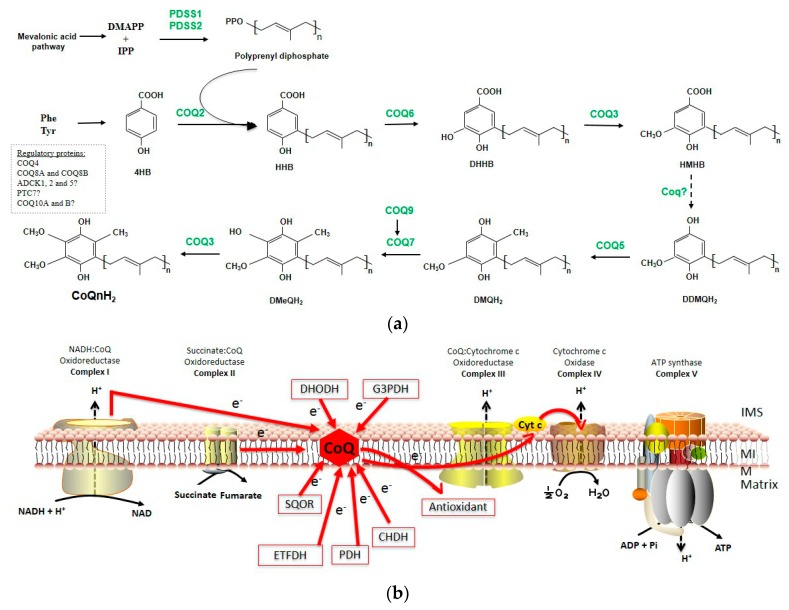
The coenzyme Q (CoQ) biosynthesis pathway and functions of CoQ. (**a**) Schematic model of mammalian cells CoQ biosynthesis pathway: 4-hydroxybenzoic acid (4HB) is the ring precursor, dimethylally pyrophosphate (DMAPP) and isopentenyl pyrophosphate (IPP) are precursors to form polyprenyl diphosphate via prenyldiphosphate synthase subunit (PDSS)1/PDSS2. CoQ2 attaches the polyisoprenyl tail to 4HB to form 3-hexaprenyl-4HB (HHB). The next intermediates are: 3-hexaprenyl-4,5-dihydroxybenzoic acid (DHHB), 3-hexaprenyl-4-hydroxy-5-methoxybenzoic acid (HMHB), 2-hexaprenyl-6-methoxy-1,4-benzenediol (DDMQH2), 2-hexaprenyl-3-methyl-6-methoxy-1,4-benzenediol (DMQH2), and 2-hexaprenyl-3-methyl-6-methoxy-1,4,5-benzenetriol (DMeQH2) to ultimately produce reduced coenzyme Q (CoQnH2). (**b**) CoQ functions in the mitochondria. CoQ accepts electrons from complex I and complex II, sulfide quinone oxidoreductase (SQOR), proline dehydrogenase 1 (PDH), coline dehydrogenase (CHDH), mitochondrial glycerol-3-phosphate dehydrogenase (G3PDH), dihydroorotate dehydrogenase (DHOH) and electron transport flavoprotein dehydrogenase (ETFDH).

**Figure 2 nutrients-11-02221-f002:**
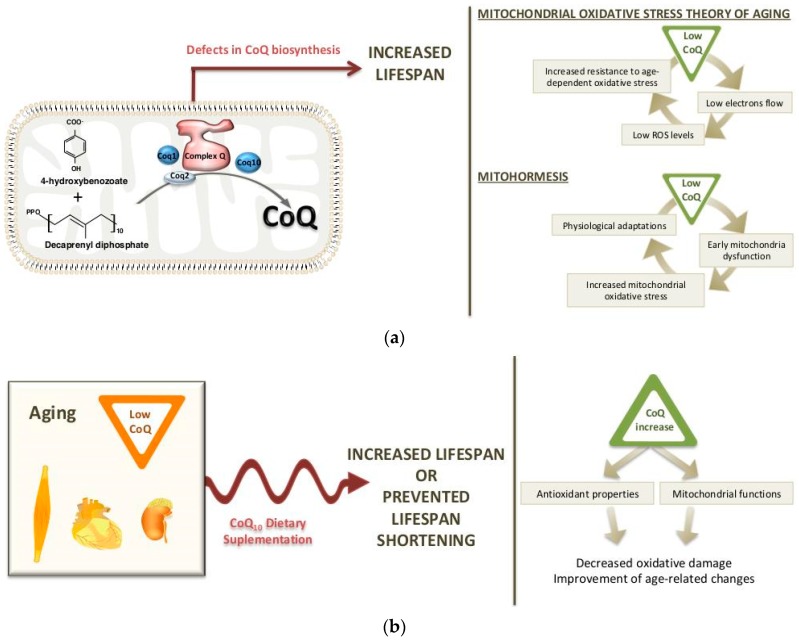
The role of coenzyme Q in aging: (**a**) Defects in CoQ biosynthesis cause decreased CoQ levels and increased life span, a fact that may be due to the changes in mitochondrial function and oxidative stress. (**b**) The increase in CoQ levels through CoQ_10_ dietary supplementation prevents some age-related damages that are associated with changes in redox states and mitochondrial function.

**Table 1 nutrients-11-02221-t001:** Coenzyme CoQ treatment effects on life span of different animal models.

Model	Strain	Age, Gender, *n*	CoQ form (Daily Dose/Conc), Treatment Duration	Diet/Food	Effect on Longevity	Ref
*Caenorhabditis elegans*	N2 Bristol wild-type	Egg, *n* = 88–105	Water-soluble CoQ_10_ ^1^ (dose)	Nematode growth medium (NGM) with *E. coli* OP50	No effect	[187]
		L1, *n* = 96–98	CoQ_10_ (50g/mL)	NGM with *E. coli* OP50	Average life span extended by 6%	[188]
			CoQ_10_ (150g/mL)	NGM with *E. coli* OP50	Average life span extended by 18%	[188]
	Clk-1 mutant	Egg, *n* = 88–105	Water-soluble CoQ_10_ ^1^ (dose), 24 h	NGM with *E. coli* OP50	No effect	[187]
		Eggs, *n* = 100	Different engineered *Escherichia coli* strains producing either CoQ_6_ to CoQ_10_	NGM with different engineered *E. coli* strains or *E. coli* OP50 (which produce CoQ_8_)	Median adult life span increased by 19% but only with CoQ_10_-producing bacteria	[118]
	Mev-1 (kn1) mutant	L1, *n* = 96–98	CoQ_10_ (50g/mL)	NGM with *E. coli* OP50	Average life span increased by 13%	[132]
			CoQ_10_ (150g/mL)	NGM with *E. coli* OP50	Average life span increased by 19%	
Mouse (*Mus musculus*)	C57BL/6	3.5 months old, *n* = 50	CoQ_10_ (93 mg/kg of bw)	ad libitum Purina diet 5001	No effect	[157]
			CoQ_10_ (371 mg/kg of bw)	ad libitum Purina diet 5001	No effect	[157]
	C57/B17	2 m old, male, *n* = 43	CoQ_10_ (10 mg/kg of bw)	normal animal diet	No effect	[170]
	C57BL/6 C3H (B6C3F1)	14 m old, male *n* = 60	CoQ_10_ (100 mg/kg)	AIN93 diet	No effect	[169]
Rat (*Rattus norvegicu*s)	Sprague–Dawley	From pregnancy, male, *n* = 75	CoQ_10_ (10 mg/kg)	normal animal diet	No effect	[170]
	Wistar	28 d old (weaning), male, *n* = 43	CoQ_10_ (2.5 mg/kg of bw)	AIN93 diet but with 8% sunflower oil or virgin olive oil as unique dietary fats	Median life span increased by 11.7%.	[180]
		28 d old (weaning), male, *n* = 22–25	CoQ_10_ (2.5 mg/kg of bw)	AIN93 diet with sunflower oil as unique dietary fat (4%)	Increased median life span by 25.5%	[171]
				AIN93 diet with fish oil as unique dietary fat (4%)	No effect	[171]
				AIN93 diet with virgin olive oil as unique dietary fat (4%)	No effect	[171]

^1^ A 122 mM emulsion with a mean diameter of 52 nm in 20% glycerol-fatty acid ester and 50% high-fructose corn syrup. Abbreviations: bw: body weight, CoQ: coenzyme Q, d: days, m: months, *n*: sample size, and h: hours.

**Table 2 nutrients-11-02221-t002:** Studies on long-term coenzyme CoQ treatment effects on age-related changes in disease-free animal models.

Model	Strain, Age, Sex	CoQ form (Dose), Duration	Diet/Food	Tissues, Organs or Systems (Sample Size Per Group)	Consequences in Age-Related Changes	Ref
*C. elegans*	N2 Bristol wild-type, Egg	Water-soluble CoQ_10_ ^1^	NGM containing 200 μg/mL streptomycin with *E. coli* OP50, 24 h	Nervous system	No effect on pharyngeal contraction and defecation rate.	[187]
	N2 Bristol wild-type, L1	CoQ_10_ (50 or 150 g/mL)	With a lawn of *E. coli* OP50	-	Reduced O_2_^−^ in the presence of succinate, although in a slight manner.	[131]
	Clk-1 mutant, Egg	Water-soluble CoQ_10_ ^1^ (0.1, 1, or 10 μM)	NGM containing 200 μg/mL streptomycin with *E. coli* OP50, 24 h	Nervous system	Increased pharyngeal pumping rate and defecation rate slowed in a dose-dependent manner, with comparable values found in wild-type strains.	[187]
		Different engineered *E. coli* strains producing CoQ_6_ to CoQ_10_	NGM with different engineered strains or the OP50 strain of *E. coli*	- (100)	Bacteria containing CoQ_6_, CoQ_7_ or CoQ_10_ decreased complex I-dependent respiration rates compared to those containing CoQ_8_ or CoQ_9_. Bacteria containing CoQ_7_ or CoQ_10_ decreased complex II-dependent respiration rates compared to those fed on bacteria containing vitamin K_12_, CoQ_6_ or CoQ_8_.	[228]
Mouse (*Mus musculus*)	C57BL/6, 3.5 m old, male,	CoQ_10_ (0.072 or 0.281%, *w*/*w*), 16/22 m of age	ad libitum Purina diet 5001	Liver, heart, skeletal muscle and brain (2–6)	No effect on enzymatic antioxidant defenses (SOD, catalase and GPX activities). No effect on protein oxidative damage (carbonyl content). No effect on mitochondrial Reactive Oxygen Species (ROS) production. No effect on glutathione redox state. No effect on mitochondrial function (mtETC complexes activities).	[157]
	C57BL/6NCr, 15/6 m old, male,	Water-soluble CoQ_10_ ^1^ (150 μM) via drinking water	Standard chow diet	Brain (motor cortex) (6–9)	Restored aging-associated decreases in mitochondrial function (OCR). Restored aging-associated motor function, phosphorylated α-synuclein and glutamate transporter 1 levels.	[229]
	SAMP1, 4 wk old, male and female	CoQ_10_ (0.2%, *w*/*w*), 10/12/14/16 m	Standard laboratory mouse diet	General (9–11)	No effect on senescence evaluated by the grading system by Hosokawa et al., 1984.	[211]
				Urine (9–11)	No effect on oxidative damage (acrolein-lysine adduct and OhdG).	
				Brain (9–11)	No effect on senile amyloid deposition rate.	[211]
		CoQ_10_ H_2_ (0.2%, *w*/*w*), 10/12/14/16 m		General (10)	Slowed senescence evaluated by the grading system.	[211]
				Urine (10)	No effect on oxidative damage (acrolein-lysine adduct and OhdG).	[211]
				Brain (10)	No effect on senile amyloid deposition rate.	[211]
	SAMP1/Sku Slc, 4 wk old, female	CoQ_10_ H_2_ (0.3%, *w*/*w*), 2/7/13/ and 19 m of age	CE-2	General (11–20)	Slowed degree of senescence ^2^. Slowed the rate of age-related hearing loss.	[210]
				Liver (11–20)	Prevented age-related decreases in the expression of sirtuin gene family members and increased intracellular cyclic AMP (cAMP) levels. Maintained mitochondrial biogenesis and oxidative metabolism by maintaining PPARγ coactivator (PGC)-1α activated. Maintained enzymatic antioxidant defenses (SOD and isocitrate dehydrogenase [IDH]2). Increased mitochondrial function (complex I activity). Decreased protein, lipid and DNA oxidative damage (protein carbonyls, MDA and apurinic/apyrimidinic sites). Increased the GSH:GSSG ratio.	[210]
Rat (*Rattus novergicus*)	Sprague–Dawley, 14 m old, male	CoQ_10_ (0.324%, *w*/*w*), 13 wk	NIH-31 diet	Blood (8)	Increased the GSH:GSSG ratio.	[156]
				Liver (8)	No effect on protein oxidative damage (protein carbonyls). No effect on enzymatic antioxidant defense (catalase, SOD and GPX).	[156]
				Heart and Brain	No effect on lipid oxidative damage (hydroperoxides). No effect on mitochondrial ROS (H_2_O_2_) production.	[156]
				Skeletal muscle	No effect on lipid oxidative damage (hydroperoxides). Decreased protein oxidative damage (protein carbonyls) at mitochondria. No effect on mitochondrial ROS (H_2_O_2_) production. No effect on enzymatic antioxidant defense (catalase, SOD and GPX).	[156]
	Wistar, 28 d old (weaning) male	CoQ_10_ (0.005%, *w*/*w*), 6/24 m of age	AIN93 diet with sunflower oil as unique fat source (4%)	Urine (6)	Reduced aging-associated increase in urinary F_2_-isoprostanes.	[174]
				Pancreas (6)	Improved endocrine pancreas structure, in particular β-cell mass.	[175]
				Bone (6)	Prevented aging-associated bone mass loss decline.	[174]
				Alveolar bone (6)	Attenuated aging-associated alveolar bone loss.	[173]
				Gingivae (6)	Increased antioxidant enzymatic defenses (antioxidant enzyme gene expression). Increased mitochondrial biogenesis markers.	[173]
			AIN93 diet with fish oil as unique fat source (4%)	PBMCs	Reduced DNA oxidative damage markers (DNA strand breaks) in 24-m-old rats.	[182]
				Urine	Reduced lipid oxidative damage markers (F2-isoprostanes) in 6-m-old rats.	[182]
				Pancreas	No effect on structural alterations in exocrine compartment.	[175]
				Bone	Increased bone mass density in 24-m-old rats.	[182]
				Alveolar bone	No effect on aging-associated alveolar bone loss.	[173]
				Gingivae	No effect on mitochondrial biogenesis markers.	[173]
			AIN93 diet with virgin olive oil as unique fat source (4%)	Urine	No effect on lipid oxidative damage (F2-isoprostanes) markers.	[174]
				Pancreas	No effect on histopathological alterations.	[175]
				Bone	No effect on aging-associated bone mass density loss.	[174]
				Alveolar bone	No effect on aging-associated alveolar bone loss.	[173]
				Gingivae	No effect on mitochondrial biogenesis markers. No effect on enzymatic antioxidant defense	[173]
		CoQ_10_ (0.005%, *w*/*w*), 6/12 m	AIN93 diet but with 8% of sunflower oil as fat source	Heart (8)	Attenuated an aging-associated increase in lipid oxidative damage (hydroperoxides).	[176]
		CoQ_10_ (0.062%, *w*/*w*), 6/12 m of age	AIN93 diet but with 8% of sunflower oil as fat source	Liver (8)	Decreased cytosolic and membrane-bound NQO1 activity.	[184]
				Brain (8)	Decreased cytosolic and membrane-bound NQO1 activity	[184]
			AIN93 diet but with 8% of virgin olive oil as fat source	Liver (8)	Decreased cytosolic and membrane-bound NQO1 activity.	[184]
				Brain (8)	Decreased cytosolic and membrane-bound NQO1 activity.	[184]
		CoQ_10_ (0.005%, *w*/*w*), 6/12/18/24 m of age	AIN93 diet but with 8% of sunflower oil as fat source	Blood (8)	Decreased DNA oxidative damage markers (DNA strand breaks) in PBMCs in 18- and 24-m-old rats.	[180]
		CoQ_10_ (0.005%, *w*/*w*), 6/12/24 m of age	AIN93 diet but with 8% of sunflower oil as fat source	Heart (20)	Decreased lipid oxidative damage (hydroperoxides) in 12- and 14-m-old rats.	[213]
				Liver (8)	Prevented an aging-associated decrease in glutathione-S-transferase (GST) activity but Se-dep GPX was not clearly affected. No effect on basal lipid oxidative damage markers (hydroperoxides) but attenuated formation against AAPH in old rats.	[183]
		CoQ_10_ (0.005%, *w*/*w*) 6/24 m of age	AIN93 diet with 8% of sunflower oil as fat source	Blood (20)	Increased non-enzymatic antioxidant defenses (α-tocopherol and retinol) and total antioxidant capacity in aged rats. Decreased DNA oxidative damage markers (DNA strand breaks) in PBMCs in young rats.	[179]
				Liver	Prevented an aging-associated increase in lipid oxidative damage markers (hydroperoxides). Increased non-enzymatic antioxidant defenses (α-Tocopherol). Prevented an aging-associated decrease in enzymatic antioxidant defenses (catalase activity).	[178]
				Skeletal muscle	Increased non-enzymatic antioxidant defenses (α-Tocopherol) in young rats but attenuated its aging-associated increase. Reduced lipid oxidative damage markers (hydroperoxides) at any age. Prevented an aging-associated increase in enzymatic antioxidant defenses (catalase activity).	[178]
				Heart	Increased non-enzymatic antioxidant defenses. Partially prevented an age-associated mitochondrial function (mtETC II and III and COX were decreased). Prevented an age-associated increase in lipid and DNA oxidative damage in mitochondria (deleted mtDNA and hydroperoxides). Improved mitochondrial ultrastructure (area, perimeter, cristae density) in aged rats.	[181]
				Brain	Increased non-enzymatic antioxidants (α-tocopherol) at mitochondria. Decreased mitochondrial ROS production. Decreased enzymatic antioxidant defenses (GPX content) in cytosol. Increased mitochondrial function (mtETC complex I, IV and III activities) in young rats, but this decreased (complex I activity) in aged rats. Decreased oxidative DNA and lipid damage markers at mitochondria (hydroperoxides and deleted mtDNA).	[178]

^1^ A 122 mM emulsion with a mean diameter of 52 nm in 20% glycerol-fatty acid ester and 50% high-fructose corn syrup. ^2^ The degree of senescence was evaluated by a grading system using eleven categories of behavioral activity and gross appearance of the skin, eyes, and spine were considered to be associated with the aging process: passivity, reactivity; glossiness, coarseness, hair loss, skin ulcers; periophthalmic lesions, corneal opacity, corneal ulcer, cataracts, and lordokyphosis. Abbreviations: AAPH: 2,2′-Azobis(2-amidinopropane) dihydrochloride, bw: body weight, cAMP: cyclic AMP, CoQ: coenzyme Q, CoQ_10_H_2_: reduced CoQ_10_, COX: cytochrome C oxidase, d: days, m: months, mETC: mitochondrial electron transport chain, n: sample size, h: hours, IDH2: isocitrate dehydrogenase 2, mtDNA: mitochondrial DNA, MDA: malondialdehyde, NQO1: NQO1-NAD(P)H dehydrogenase [quinone] 1 reductase, OCR: oxygen consumption rate; OhdG; 8-hydroxydeoxyguanosine; PBMCs: peripheral blood mononuclear cells, GPX: Glutathione peroxidase; GSH: reduced glutathione, GSSG: oxidized glutathione, GST: glutathione-S-transferase, ROS: reactive oxygen species, Se-dep: selenium-dependent, and SOD: superoxide dismutase.

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
