# Peer review of "The Paradox of Coenzyme Q10 in Aging"

_nutrients, 2019, doi:10.3390/nu11092221_

Round 1

Reviewer 1 Report

The review by Diaz-Casado aims to approach the difficult concept of Coenzyme Q10 and ageing. In general, the review is exhaustive and nicely describes the at times confusing and inconsistent literature of this topic.

The initial description on CoQ10 synthesis is a nice review in itself and not necessarily novel (and could therefore be shortened to provide more space for the topic of the review based on its title) but sets the scene for the following chapters.

Major points:

One of the major points only partially addressed in the first section is that rodents do not use CoQ10 as the main quinone. Given the intricate interaction between CoQ and its mitochondrial binding site, the availability of a different than physiologically used CoQ has to be regarded with extreme caution.  Therefore supplementation with CoQ10 can only provide limited information with regards to mitochondrial function and this should be highlighted and perhaps addressed in a separate chapter.

The authors describe that mito-hormesis is unlikely the cause for extended life in the COQQ95X mouse (line 397). This is surprising and unfortunately one paper on the topic (Rodrques-Hidalgo 2018) did not measure lipid peroxidation in the mouse model which would be the most likely oxidative damage (see minor comment below), nor mitochondrial superoxide production. Measuring ATP production is typically a bad marker for the mild mitochondrial dysfunction expected in the mito-hormesis paradigm, which is supported by the prior publication from the same group (Luna-Sanchesz 2015). So it can currently not being ruled out that mito-hormesis is present in this model based on the published evidence.

When looking at human trials it would be important to point out in the text (or a table) what absorption levels actually were detected in patients in the clinical trials, how they were confirmed and whether there was a correlation between plasma levels and effect size. Given that CoQ10 absorption is notoriously variable, this has to be assessed to get some sort of evidence base.

The information that CoQ10 effectiveness is dependent on the type of oil used in a study, supports the importance of measuring plasma levels since oil will significantly help absorption of CoQ10 via bile acid production.

The statement (line 521-522) has to be retracted as the authors of the original reference clearly stated that only 1 out of 6 studies found an effect, while 2 studies were classified as biased. In addition, another Cochrance review by Ho et al found no evidence for any effectiveness of CoQ10 on blood pressure.  

In the conclusions the authors associate mito-hormesis with decreased oxidative stress (line 618), which is clearly wrong (and most likely an oversight). The initial mito-hormesis theory explicitely suggests increased ROS as the signal of cellular adaptation (see Tapia et al. 2006, Schulz et al. 2007 as well as Ristow et al. 2009), which is not age-dependent and can clearly be initiated for example in the muscle by exercise in adult individuals.

I would encourage the authors to be more critical around the available evidence. While the text provides evidence of more negative or unclear results than actual positive results of CoQ10 supplementation, the conclusion states that “ CoQ10 supplementation has shown therapeutic benefits in a wide variety of… studies……”. The text of this review does not support this statement and large adequately-powered human studies are still missing to unequivocally prove therapeutic benefits. Therefore, it appears that this is something the authors are passionate about and want to push but which is not reflected by the current state of published research.

Minor comments:

I would strongly suggest that the authors get a native speaker to edit the review due to frequent grammatical and typographical errors.

Line 210: Reference missing for statement. Also CoQ10 is absolutely restricted to the membrane compartment due to its extreme lipophilicity. Hence it is very difficult to reconcile how it could prevent protein damage (unless also restricted to the membrane) or DNA damage. Citation of an old reference (71) is not helpful to support this point and should be replaced by recent primary evidence (instead of citing a review). If this can not be found, I would suggest to delete that statement. If sufficient published evidence for this can be found, the authors should attempt to discuss how this effect could be mechanistically induced in molecular detail.

Author Response

We are grateful for the reviewer's efforts in reviewing our manuscript, as their criticisms have been very constructive and have helped improve our previous work.

Major points:

Reviewer: One of the major points only partially addressed in the first section is that rodents do not use CoQ10 as the main quinone. Given the intricate interaction between CoQ and its mitochondrial binding site, the availability of a different than physiologically used CoQ has to be regarded with extreme caution.  Therefore supplementation with CoQ10 can only provide limited information with regards to mitochondrial function and this should be highlighted and perhaps addressed in a separate chapter.

Authors: We agree with the reviewer in the fact that the size of the isoprenoid chain may be relevant for the biological activity of CoQ. However, it is unclear why human and rodents have both CoQ9 and CoQ10 and why the distribution of both CoQ forms differ between tissues.  

Reviewer: The authors describe that mito-hormesis is unlikely the cause for extended life in the COQQ95X mouse (line 397). This is surprising and unfortunately one paper on the topic (Rodrques-Hidalgo 2018) did not measure lipid peroxidation in the mouse model which would be the most likely oxidative damage (see minor comment below), nor mitochondrial superoxide production. Measuring ATP production is typically a bad marker for the mild mitochondrial dysfunction expected in the mito-hormesis paradigm, which is supported by the prior publication from the same group (Luna-Sanchesz 2015). So it can currently not being ruled out that mito-hormesis is present in this model based on the published evidence.

Authors: Rodriguez-Hidalgo 2018 explain that the extension of life expectancy in the COQQ95X mouse is not due to early mitochondrial dysfunction in the liver. The authors did not find differences in the mitochondrial bioenergetics (measuring mitochondrial respiratory chain activities), oxidative damage and antioxidant enzymes in the liver of Coq9Q95X . Then the authors proposed that the increase in life expectancy could be due to mitochondrial dysfunction in other tissues but not due to hepatic mitohormesis. We do not rule out in the paragraph that mitohormesis is the mechanism, but we cannot discard it and further investigation is needed. We have introduced a clarification in text (Page 11, lines 423 to 428 ):

“Unlike mclk1+/- mice, Coq9Q95X mice have neither decrease of the CoQ levels in liver, DMQ accumulation nor mitochondrial dysfunction in this tissue. Therefore, there is no evidence that the increased lifespan in Coq9Q95X mice is due to early mitochondrial dysfunction in liver, as it was proposed in mclk1+/- mice. Although, the increased lifespan in Coq9Q95X mice could be attributed to a mitochondrial dysfunction in other tissues, as brain, kidney or muscle, or to unknown mechanisms [138].”

Reviewer: When looking at human trials it would be important to point out in the text (or a table) what absorption levels actually were detected in patients in the clinical trials, how they were confirmed and whether there was a correlation between plasma levels and effect size. Given that CoQ10 absorption is notoriously variable, this has to be assessed to get some sort of evidence base.

Authors: We agree with the reviewer in the importance that confirming absorption and incorporation of CoQ is very relevant to explain possible differences in the effects of this type of interventions. For this reason, we have introduced this information along the text when it was available. Unfortunately, some studies did not provided information about this issue and in many cases this did not explain why there are strongest effects or not.

Reviewer: The information that CoQ10 effectiveness is dependent on the type of oil used in a study, supports the importance of measuring plasma levels since oil will significantly help absorption of CoQ10 via bile acid production.

Authors: The proposal of the effect of CoQ10 supplementation could be conditioned by the different oils present in the diet since they help absorption of CoQ10 via bile acid production is very interesting. For this reason, data of CoQ levels presented in publications comparing effects of diet with different oil have been included and an alternative proposal for explaining CoQ different effect have been included now in the text as follows: (Page 12, lines 507 to 513):

“On the other hand, the different oils used could modulate CoQ10 absortion since fatty acids significantly help absorption of CoQ10 via bile acid production [185,186]. In animals fed on similar CoQ-supplemented diets, plasma CoQ10 concentration values were 39.24 ± 9.25 and 50.71 ± 5.22 μM when they recceived a diet based on virgin olive oil and sunflower oil, repectively [174]. Other study reported even lower values in those animals fed on a fish oil based diet [182]. This seems in concordance with the proposed hypothesis but the values found in animals receiving different oil were not statistically compared in these studies”

Reviewer: The statement (line 521-522) has to be retracted as the authors of the original reference clearly stated that only 1 out of 6 studies found an effect, while 2 studies were classified as biased. In addition, another Cochrance review by Ho et al found no evidence for any effectiveness of CoQ10 on blood pressure.

Authors: We thank the reviewer for its attention noting this error. We have reviewed the reference and we agree with the reviewed, the statement has to be retracted. This has been modified as follows (Page 15, lines 560 to 562):

“This systemic review does not support efficacy of CoQ10 supplements in reducing systolic blood pressure, total cholesterol, LDL- C, or HDL-C”

Reviewer: In the conclusions the authors associate mito-hormesis with decreased oxidative stress (line 618), which is clearly wrong (and most likely an oversight). The initial mito-hormesis theory explicitely suggests increased ROS as the signal of cellular adaptation (see Tapia et al. 2006, Schulz et al. 2007 as well as Ristow et al. 2009), which is not age-dependent and can clearly be initiated for example in the muscle by exercise in adult individuals.

Authors: We thank the reviewer for noting this error, which has been modified in the paragraph as follow (page 27, lines 690 to 694):

“Some defects in CoQ biosynthesis induce an increase in lifespan in animal models, most likely due to either a mild reduction in the flow of electrons in the mitochondrial respiratory chain and the subsequent decrease in ROS production or an early mitochondrial dysfunction and a physiological adaptation to this condition, the latter phenomenon being compatible with the concept of mitohormesis.”

Reviewer: I would encourage the authors to be more critical around the available evidence. While the text provides evidence of more negative or unclear results than actual positive results of CoQ10 supplementation, the conclusion states that “ CoQ10 supplementation has shown therapeutic benefits in a wide variety of… studies……”. The text of this review does not support this statement and large adequately-powered human studies are still missing to unequivocally prove therapeutic benefits. Therefore, it appears that this is something the authors are passionate about and want to push but which is not reflected by the current state of published research.

Authors: We thanks to the reviwer for pointing this issue. We agree with the fact of there are not too evidence supporting therapeutic benefits of CoQ10 since all studies did not found benefits. For this reason conclusion has been modified to be more accurately with the reality.

Minor comments:

Reviewer: I would strongly suggest that the authors get a native speaker to edit the review due to frequent grammatical and typographical errors.

Authors: The text have been reviewed.

Reviewer: Line 210: Reference missing for statement. Also CoQ10 is absolutely restricted to the membrane compartment due to its extreme lipophilicity. Hence it is very difficult to reconcile how it could prevent protein damage (unless also restricted to the membrane) or DNA damage. Citation of an old reference (71) is not helpful to support this point and should be replaced by recent primary evidence (instead of citing a review). If this can not be found, I would suggest to delete that statement. If sufficient published evidence for this can be found, the authors should attempt to discuss how this effect could be mechanistically induced in molecular detail.

Authors: We have added two references for this statement. The reference 71 has been changed by the original article Forsmark-Andree, P. 1990. We have added two studies, Tomasetti, M 1999 and Tomasetti, M 2001, for supporting this statement. We discuss the mechanisms that could be involved in the protective role of CoQ against oxidative damage to DNA. The paragraph has been changed as follows (page 6, lines 227 to 235):

“CoQ is effective in preventing DNA damage by decreasing the strand breaks in DNA and eliminating the oxidative DNA damage in mitochondria [73], which is important for mitochondrial DNA since its oxidative damage is about 10-fold higher than nuclear DNA and is not easily reparable. These data are supported by Tomasetti and colleagues [74,75] who showed that the CoQ10 supplementation inhibits strand breaks DNA formation and oxidative DNA damage, and enhances DNA repair enzyme activity in lymphocyte. The antioxidant activity of CoQ10 is able to inhibition additional oxidative damage in the cells or the role that CoQ10 may play in the redox mechanism implicated in trans-activation of DNA repair enzymes could explain the capability of CoQ10 to protect DNA form oxidative damage. However, the precise mechanism remains unclear.”

Reviewer 2 Report

This review covered recent progress and previous works of several different aspects of Coenzyme Q including biosynthesis, its roles, deficiency, in addition to ageing which they focus on. The effect of CoQ10 on ageing is still the matter of debate. Overall, I think this review is well summarized.

Minor points are listed below.

Line 46; It was proposed that CoQ is also synthesized in Golgi in addition to mitochondria, but considering all CoQ biosynthetic enzymes reside in mitochondria and current studies showed that they are concentrated on ERMES (Eisenberg-Bord et al., Contact 2019), the earlier finding that CoQ is present in Golgi does not necessary mean it is synthesized in Golgi, but I think it is transferred to Golgi from ERMES.

Line 53; To be more accurate, PDSS1 and PDSS2 are not so called orthologues of COQ1. These two proteins are subunits of one enzyme required for the activity of prenyl diphosphate synthase (Saiki et al. FEBS J 2005). PDSS1 or PDSS2 alone is not functional, whereas COQ1 alone is functional as an enzyme.

Line 55; The reference 10 does not focus on the conservation of human COQ orthologs in yeast. Over all conservation is described in (Hayashi et al., PloS One 2014)

Line 94; ‘decarboxylations’ must be single ‘decarboxylation’

Line 111; Ref [29] does not focus on CoQ. ‘White MD et al., Nature 2015’ is appropriate

Line 164; I think the argument about potential toxicity of intermediate products are not so solid in the cited references [54,55].  

Line 180; Initial finding of the relevance of sulfide and CoQ is much early described in Zhang et al., Biofactor 2008

Line 185; [62,63] [64] >> [62,63,64]

Line 187; hydroxi >> hydroxyl

Line 243; I feel that citing 12 references [82-93] without any specific explanation of individual reference is not appropriate. If the authors are not focusing on this aspect, they can only cite the representative reference.

Line 331; Reference 128 describes the failure of DMQ to support respiration, so it is an opposite evidence from the stated meaning.

Line 471; The same problem I pointed out above is also here on the citation [146-154]. Many citation, but not clearly stated the difference of cited works.

Line 571; could be also occur >> could also occur

Lines 932, 945, and 953;  CoQ(1)(0) ---  They need to be fixed.

Lines 1267-1278; duplication of numbers need to be fixed.

Author Response

The authors are grateful for the reviewer's efforts in reviewing our manuscript, as their criticisms have been very constructive and have helped improve our previous work.

Minor points are listed below.

Reviewer: Line 46; It was proposed that CoQ is also synthesized in Golgi in addition to mitochondria, but considering all CoQ biosynthetic enzymes reside in mitochondria and current studies showed that they are concentrated on ERMES (Eisenberg-Bord et al., Contact 2019), the earlier finding that CoQ is present in Golgi does not necessary mean it is synthesized in Golgi, but I think it is transferred to Golgi from ERMES.

Authors: Eisenberg-Bord show that there is a relationship between ERMES, CoQ biosynthesis and the integrity of the Q complex, so a reduction in ERMES alters the biosynthesis of CoQ6 in mitochondria and affects mitochondrial respiration, as well as the the transport of CoQ, intermediates and precursors. So the statement that CoQ could reach the Golgi from the mitochondria by ERMES is an assumption that is not enoughly suppoted by experimental data yet. But considering the suggestion we have modified the paragraph as follows (Page 2, lines 46 to 52):

 “The synthesis of CoQ in eukaryotes mainly occurs in mitochondria by a set of nuclear-encoded COQ proteins through a biochemical pathway that has not been fully defined. Moreover some authors have suggested that CoQ may be also produced outside of mitochondria, mainly in the endoplasmic reticulum-Golgi [1,2]. On the other hand, recent studies suggest that the endoplasmic reticulum mitochondria encounter structure (ERMES) might promote the CoQ biosynthesis and the movement of CoQ and its precursors and intermediates between these organelles [3]. Thus, mitochondrial production of CoQ could contribute to extramitochondrial pools of CoQ..”

Reviewer: Line 53; To be more accurate, PDSS1 and PDSS2 are not so called orthologues of COQ1. These two proteins are subunits of one enzyme required for the activity of prenyl diphosphate synthase (Saiki et al. FEBS J 2005). PDSS1 or PDSS2 alone is not functional, whereas COQ1 alone is functional as an enzyme.

Authors: We agree with the referee and we have made the correction in the text and incorporated the reference as follows (Page 2, lines 59 to 61):

In the cases COQ8 and COQ10, human have two orthologues for each one: ADCK3 (COQ8A) and ADCK4 (COQ8B), and COQ10A and COQ10B, respectively. In H. sapiens the polypernyl diphosphate synthase is heterotetramer formed by dimer of PDSS1 and PDSS2 proteins [10,11].”

Reviewer: Line 55; The reference 10 does not focus on the conservation of human COQ orthologs in yeast. Over all conservation is described in (Hayashi et al., PloS One 2014)

Authors: We agree and we have changed the reference 10 by Hayashi et al., 2014, now is the reference 12.

Reviewer: Line 94; ‘decarboxylations’ must be single ‘decarboxylation’

Authors: We have modified it in the text (line 130)

Reviewer: Line 111; Ref [29] does not focus on CoQ. ‘White MD et al., Nature 2015’ is appropriate

Authors: We agree and we have changed the reference 29 by White et al., 2015, now is the reference 32.

Reviewer: Line 164; I think the argument about potential toxicity of intermediate products are not so solid in the cited references [54,55].  

Authors: We have modified these references: Stefely, J. 2016, Lohman, D.C. 2014 and Allan, C, 2015.

Reviewer: Line 180; Initial finding of the relevance of sulfide and CoQ is much early described in Zhang et al., Biofactor 2008

Authors: We have added this reference, Zhang et al., 2008 is the reference 61.

Reviewer: Line 185; [62,63] [64] >> [62,63,64]

Authors: We have modified it in the text and due to the changes made in the bibliography they now correspond to the references [64-66].

Reviewer: Line 187; hydroxi >> hydroxyl

Authors: Done

Reviewer: Line 243; I feel that citing 12 references [82-93] without any specific explanation of individual reference is not appropriate. If the authors are not focusing on this aspect, they can only cite the representative reference.

Authors: The large number of references is due to the citation of publications referring to mutations in 10 different genes that cause primary deficiencies in CoQ10. We have modified and introduced each reference after each gene. We think it is important to give references for each of the genes that have been published and diagnosed to date. The paragraph has been modified as follows (Page 7, lines 263 to 269):

 “Since then, approximately 200 patients have been described with pathogenic mutations in PDSS1 [88], PDSS2 [89, 90], COQ2 [88, 89, 91], COQ6 [89, 92], COQ7 [22], COQ4 [93-95], COQ5 [96], COQ8A [36-38], COQ8B [89, 97] and COQ9 [98] genes. The clinical manifestations are very heterogeneous among different genes and among patients with mutations in individual genes. Multiple organs may be affected and lead to different symptomatology; when the central nervous system (CNS) is affected the clinical manifestations are variable, including encephalopathy, seizures, cerebellar ataxia, epilepsy, intellectual disability, hypotonia, dystonia, spasticity; in kidney the most common manifestations is steroid-resistant nephrotic syndrome (SRNS); myopathy in muscle and hypertrophic cardiomyopathy in heart. Many symptoms are common to other mitochondrial diseases [99].”  

Reviewer: Line 331; Reference 128 describes the failure of DMQ to support respiration, so it is an opposite evidence from the stated meaning.

Authors: We agree with the reviewer, we made a mistake. The reference Padilla, S 2004 has been deleted from the paragraph.

Reviewer: Line 471; The same problem I pointed out above is also here on the citation [146-154]. Many citation, but not clearly stated the difference of cited works.

Authors: the number of citations have been reduced for including only the most relevant studies.

Reviewer: Line 571; could be also occur >> could also occur

Authors: Done

Reviewer: Lines 932, 945, and 953;  CoQ(1)(0) ---  They need to be fixed.

Authors: Done

Reviewer: Lines 1267-1278; duplication of numbers need to be fixed.

Authors: Done

Reviewer 3 Report

This is a very interesting, intelligent and well written summery of the current knowledge on CoQ10 and aging. 

Comments:

The main subject of the article is the paradox between the seemingly contradicting findings of improved longevity with both mutations that reduce CoQ9 / CoQ10 production as well as with external administration of CoQ10. The authors provide information about the paradox but do not suggest  a comprehensive explanation. One such explanation would be that the endogenous production of CoQ10 may be somehow detrimental to the mitochondria or the cell and if replaced by an external source such as dietary CoQ10 or CoQ10 producing bacteria it provides its needed function while avoiding the harmful effects of its production. The section about the bio-synthesis of CoQ10 and its action would be better explained with an illustration. Although this paper did not aim to cover the subject of reproductive senescence, this is an interesting aspect of the process of aging that had been studied extensively for the effects of CoQ10 and of its deficiency and should be included. Line 405, please provide reference to dietary CoQ10 cellular incorporation. Is there any evidence that dietary CoQ10 enters the mitochondria. Evidence supporting CoQ10 mitochondrial uptake or against it would provide valuable insight on the mechanism of action of supplemental CoQ10. 

Author Response

The authors are grateful for the reviewer's efforts in reviewing our manuscript, as their criticisms have been very constructive and have helped improve our previous work.

Comments:

Reviewer: The main subject of the article is the paradox between the seemingly contradicting findings of improved longevity with both mutations that reduce CoQ9 / CoQ10 production as well as with external administration of CoQ10. The authors provide information about the paradox but do not suggest a comprehensive explanation. One such explanation would be that the endogenous production of CoQ10 may be somehow detrimental to the mitochondria or the cell and if replaced by an external source such as dietary CoQ10 or CoQ10 producing bacteria it provides its needed function while avoiding the harmful effects of its production.

Authors: Since there are still many icognites regarding all the roles of coenzyme Q and even of the possible intermediaries involved in their syntheses, it is very complicated and risky to be able to offer a global explanation to the paradox presented here, we only have partial explanations. On the one hand there are certain models with deficit in the production of CoQ that carry disease, although certain models with deficiencies have shown high longevity, for which different hypotheses have been established highlighting that of mitohormesis. On the other hand, the contribution of Coenzyme Q has shown positive effects on health or has increased longevity only in specific cases. Therefore, it seems that the conditions already present in which treatment is initiated, in many cases related to deficiencies of this molecule and / or oxidative stress and mitochondrial involvement, or with the dietary context, situations under which CoQ contribution is more useful. In addition, these can also influence the absorption and uptake of CoQ, making it more bioavailable. However, in many cases there are no data to support this since either or not body levels have been measured or only in blood. However, we have modified the conclusion because it implied that the exogenous contribution of CoQ practically always increases longevity or improves health and there are also cases where this effect is not observed.

Reviewer: The section about the bio-synthesis of CoQ10 and its action would be better explained with an illustration.

Authors: a new figure about this issue has been added to the manuscript.

Reviewer: Although this paper did not aim to cover the subject of reproductive senescence, this is an interesting aspect of the process of aging that had been studied extensively for the effects of CoQ10 and of its deficiency and should be included.

Authors: We agree with the reviewer about the potential interest of CoQ against reproductive senescence. For this reason we have included a new paragraph focused on this issue in section 3.3. Reversal of age-related changes by CoQ10 supplementation:

“Finally, there are also some studies supporting the possible use of CoQ10 agaisnt the decline in reproductive success with increasing age (i.e. reproductive senescence) and particularly against fertility decline. Female reproductive capacity declines with age as consequence of an age-related decrease in oocyte quality and quantity, which is accompanied by mitochondrial dysfunction associated with decreased oxidative phosphorylation and reduced ATP level [222-224]. Moreover, there are an increase in meiotic spindle abnormalities favouring oocyte aneuploidy that leads to a reduced embryo quality, as well as an increased incidence of miscarriages and birth defects [225]. It has been proposed that the impaired mitochondrial performance associated to reproductive senescence could be a consequence of a suboptimal CoQ10 availability that will lead to age-associated oocyte deficits [223]. Actually, it has been reported a diminished expression of genes encoding two enzymes responsible for CoQ production, Pdss2 and Coq6 in oocytes of older females in both mouse and human [223]. Likewise, oocyte-specific Pdss2-deficient animals also shown a diminished ovarian reserve, a condition prevented by maternal dietary administration of CoQ10  [223]. A similar  intervention also reversed age-related decline in oocyte quality and quantity in an aging mouse model [223]. In addition, cumulus granulosa cells also display a decline in cell function linked to mitochondrial activity accompanied by a decreased expression of Pdss2 and CoQ6 genes in, both, human and mouse model [225]. Concerning these cells, supplementation with CoQ10 restored cumulus cell number, stimulated glucose uptake, and increased progesterone production in an aged mouse model. These preclinical studies suggests that CoQ10 might improve fertility in females of advanced maternal age by improving the mitochondrial metabolism in oocyte and cumulus cells which quantity and quality would result increased [225]. However, there are no clinical studies in ageg women confirming this. On the other hans, male infertility hasa also been associated with oxidative stress and healthy sperm has been correlated with CoQ10 levels in seminal fluid [226]. In addtion, a meta-analysis of 15 randomized clinical trials suggests that some dietary CoQ10 could beneficially modulate sperm quality parameters improving male fertility. Namely, these interventions have shown to increase total sperm concentrations, sperm count and enhance sperm total motility was enhanced by CoQ10 [227]. Therefore, dietary CoQ10 also could result usefull to reduce male fertility decline. However, the limited sample size of the meta-analyzed studies and the interstudy heterogeneity are important factors for carefully interpret these results [227].”

Reviewer: Line 405, please provide reference to dietary CoQ10 cellular incorporation. Is there any evidence that dietary CoQ10 enters the mitochondria. Evidence supporting CoQ10 mitochondrial uptake or against it would provide valuable insight on the mechanism of action of supplemental CoQ10. 

Authors: there are studies finding an increase of CoQ10 levels in cell but also in mitochondrial membranes of animals supplemented with this molecule. A new reference reviewing these findings has been added to this line.